# Autocatalytic amplification of Alzheimer-associated Aβ42 peptide aggregation in human cerebrospinal fluid

Rebecca Frankel[1,10]*, Mattias Törnquist [1,10]*, Georg Meisl [2], Oskar Hansson[3,4], Ulf Andreasson[5,6], Henrik Zetterberg[5,6,7,8], Kaj Blennow[5,6], Birgitta Frohm[1], Tommy Cedervall[1], Tuomas P.J. Knowles[2,9], Thom Leiding[1] & Sara Linse [1]*

Alzheimer's disease is linked to amyloid β (Aβ) peptide aggregation in the brain, and a detailed understanding of the molecular mechanism of Aβ aggregation may lead to improved diagnostics and therapeutics. While previous studies have been performed in pure buffer, we approach the mechanism in vivo using cerebrospinal fluid (CSF). We investigated the aggregation mechanism of Aβ42 in human CSF through kinetic experiments at several Aβ42 monomer concentrations (0.8–10 μM). The data were subjected to global kinetic analysis and found consistent with an aggregation mechanism involving secondary nucleation of monomers on the fibril surface. A mechanism only including primary nucleation was ruled out. We find that the aggregation process is composed of the same microscopic steps in CSF as in pure buffer, but the rate constant of secondary nucleation is decreased. Most importantly, the autocatalytic amplification of aggregate number through catalysis on the fibril surface is prevalent also in CSF.

[1] Department of Biochemistry and Structural Biology, Lund University, P O Box 124SE22100 Lund, Sweden. [2] Department of Chemistry, Cambridge University, Lensfield Road, Cambridge CB2 1EW, UK. [3] Clinical Memory Research Unit, Department of Clinical Sciences, Lund University, Lund, Sweden. [4] Memory Clinic, Skåne University Hospital, Malmö, Sweden. [5] Department of Psychiatry and Neurochemistry, Institute of Neuroscience and Physiology, the Sahlgrenska Academy at the University of Gothenburg, Mölndal, Sweden. [6] Clinical Neurochemistry Laboratory, Sahlgrenska University Hospital, Mölndal, Sweden. [7] Department of Neurodegenerative Disease, University College London Institute of Neurology, Queen Square, London, UK. [8] UK Dementia Research Institute at University College London, London, UK. [9] Department of Physics, Cavendish Laboratory, University of Cambridge, Cambridge CB3 0HE, UK. [10]These authors contributed equally: Rebecca Frankel, Mattias Törnquist. *email: rebecca.frankel@biochemistry.lu.se; mattias.tornquist@biochemistry.lu.se; sara.linse@biochemistry.lu.se

Alzheimer's disease (AD) is a devastating neurodegenerative disease, which affects a large and increasing number of individuals world-wide[1,2]. The sequence of events leading to the initiation and propagation of this neurodegenerative disease is still largely unknown; however, in addition to aging and the major genetic risk gene apolipoprotein E (*APOE*), possible mechanisms including inflammation[3], altered endosomal sorting[4,5], and tau spreading[6] are gaining increasing awareness. There is ample evidence linking the self-assembly of the amyloid β peptide (Aβ) to the disease[7,8]. According to the amyloid cascade hypothesis, Aβ self-assembly is followed by truncation, phosphorylation and aggregation of tau, and subsequent neuronal death[9–11].

Aβ was discovered in 1984[12] and its complete amino acid sequence in plaques was determined in 1985[13]. Several length variants of this peptide have been found to coexist in body fluids, including cerebrospinal fluid (CSF) and blood serum[14–17]. Typical Aβ42 concentration in CSF are around 250 pM in healthy humans, with a large variation between individuals. Around 50% lower average Aβ42 concentration is measured in CSF from AD patients, which is thought to arise from sequestration of Aβ42 into plaques in the brain[18]. Evidence for the association between Aβ and AD largely come from the early onset of AD-type pathology in individuals with extra dosage due to chromosome 21 trisomy[19,20], as well as from familial AD cases with mutations close to the γ secretase cleavage site of amyloid precursor protein (APP)[21,22] and before the β secretase cleavage site at the N-terminus of Aβ[23], or with Aβ variants that are more aggregation-prone or follow an altered aggregation pathway[24–27].

Self-assembly of Aβ leads to the formation of elongated fibrillar aggregates of highly ordered structure[28,29], whereas smaller aggregates—oligomers—formed during the process seem to be responsible for the emergence of neurotoxicity[30]. Oligomers are transient species and a range of definitions has been used in the literature, including aggregates with lower growth rate than fibrils, aggregates in a discrete size range, or aggregates that are toxic, as reviewed[31]. One route towards future diagnostics and treatment of AD involves finding the mechanism of Aβ aggregation in terms of the microscopic steps in the process[32] and the molecular driving forces of each step[26,33]. Such knowledge can be used to search for inhibitors of particular steps, most importantly those steps leading to toxicity[34,35].

Recent investigations of highly purified peptide in buffer have revealed a mechanism that is compatible both with a large set of aggregation kinetics data at multiple peptide concentrations and with experiments using isotope-enrichment to decipher the origin of oligomeric species[32]. The result is a double nucleation mechanism compatible with the aggregation of Aβ42[32], Aβ40[36] as well as several point mutations[26,27] and variants with N-terminal extensions[37]. Primary nucleation in solution is accompanied with a very high energy barrier compared to elongation of existing aggregates[38]. Once fibrillar aggregates have formed, these present a catalytic surface for secondary nucleation of monomers. This secondary nucleation process is associated with a much lower energy barrier than primary nucleation[38] and leads to an autocatalytic process with rapid multiplication of the number of aggregates. Although secondary nucleation of Aβ monomers on fibril surface was not discovered until 2013, this type of process, leading to autocatalytic amplification of aggregate mass, has been known in crystallization and other self-assembly processes for more than a century, as reviewed[39–42].

Aβ aggregation in vivo occurs in complex fluids containing many thousands types of proteins in addition to phospholipids, metabolites, salts, and other components. At least two approaches are possible to understand any changes in the mechanism compared to the pure buffer system. In a bottom-up approach, components from relevant fluids are added one at a time and the effect on the Aβ aggregation mechanism is quantified to find the effects on each of the underlying microscopic steps. This approach is the fastest route towards a detailed physico-chemical understanding of the effects from each component and was recently used to understand the effects of ionic strength[33,43] and pH[26]. In a top-down approach, as in the current work, Aβ aggregation is studied in a full body fluid. This approach is the fastest route towards the aggregation mechanism in a complex environment, but does not provide a clear understanding of the connection between effects and effectors due to multiple simultaneous effects.

In this study, we investigate the aggregation of Aβ42 in the presence of CSF using the amyloid-binding dye thioflavin T (ThT), which, when used in a fluorescence assay, produces remarkably reproducible data over a range of peptide concentrations in a pure buffer system[44]. We also use cryo-EM to confirm that the end stage consists of fibrils very similar to those seen in pure buffer systems. The kinetic data generated by the ThT assay remains reproducible and amenable to kinetic analysis over a wide span of CSF concentrations. Pure monomer, or pure monomer supplemented with pre-formed seeds, was combined with CSF at multiple peptide concentrations and a model was globally fitted to the data as a function of concentration and time[45] to find the minimal mechanism compatible with all data. Our conclusion is that although CSF has a generally retarding effect on Aβ aggregation, the same mechanism, including a double step secondary nucleation process, remains dominant in the presence of CSF.

## Results

**Validation of automated pipetting by robot.** The pipetting robot used for all dilution series in this work was validated by dispensing different concentrations of a fluorescent dye solution and measuring the resulting fluorescent signals. The same concentration was dispensed four times within each experiment and three separate experiments were made. The results show a high degree of linearity between expected concentration and fluorescence intensity and high reproducibility both within and between the independent experiments (Supplementary Fig. 1).

**Aβ42 aggregation kinetics in buffer.** Since CSF contains ca. 140 mM NaCl and 1 mM CaCl$_2$[46], the aggregation kinetics starting from a range of Aβ42 monomer concentrations, 0.8–10 μM, were first studied in 20 mM HEPES/NaOH, 140 mM NaCl, 1 mM CaCl$_2$, pH 8.0 with 10 μM ThT (due to the addition of calcium, use of phosphate buffer as in our previous work was not possible). Four separately mixed replicates of each solution were placed in separate wells of a 96-well plate (Corning 3881 PEGylated polystyrene black plates with half-area wells with clear bottom) and the ThT fluorescence measured through the bottom of the plate. The whole experiment was repeated twice. The aggregation curves from one experiment are shown in Fig. 1a and from the other in Supplementary Fig. 2. The time at which the ThT intensity has reached half-way in between the initial baseline and the final plateau values, $t_{1/2}$, was extracted from each curve and plotted versus Aβ42 concentration in Fig. 1b, c, and a power function was fitted to the resulting values

$$t_{1/2} = \alpha \cdot c^{\gamma} \tag{1}$$

where $\gamma$ is the scaling exponent, $c$ the Aβ42 concentration, and $\alpha$ a proportionality constant. We obtain an overall average of $\gamma = -0.3$ for the two experiments shown in Fig. 1a and Supplementary Fig. 2, while $\gamma = -0.8$ was recently found in phosphate buffer

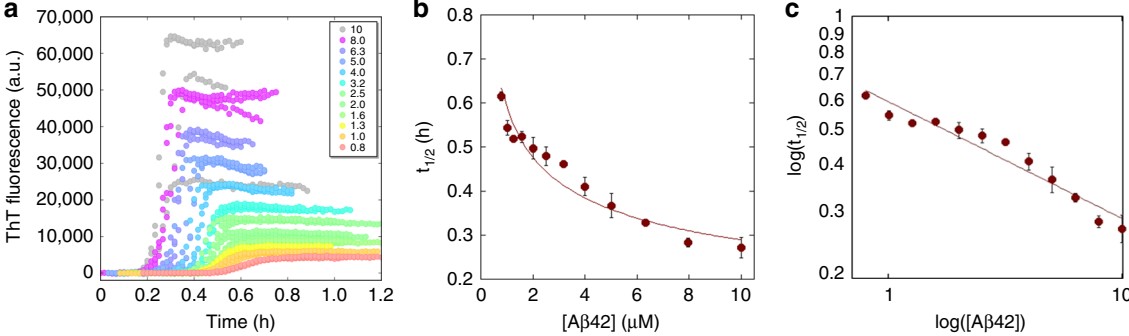

**Fig. 1 Aggregation kinetics in buffer. a** The ThT fluorescence intensity as a function of time (h) for 0.8–10 μM Aβ42 in buffer. **b** The calculated $t_{1/2}$ (h) as a function of Aβ42 concentration, with a fitted power function as in Eq. (1). The error bars are the standard deviations from the $n = 4$ repeats per Aβ42 concentration. **c** Shows the double log representation of (**b**). The fluorescence data underlying each time trace as well as the $t_{1/2}$ values and the associated standard deviations can be found in Supplementary Data 1

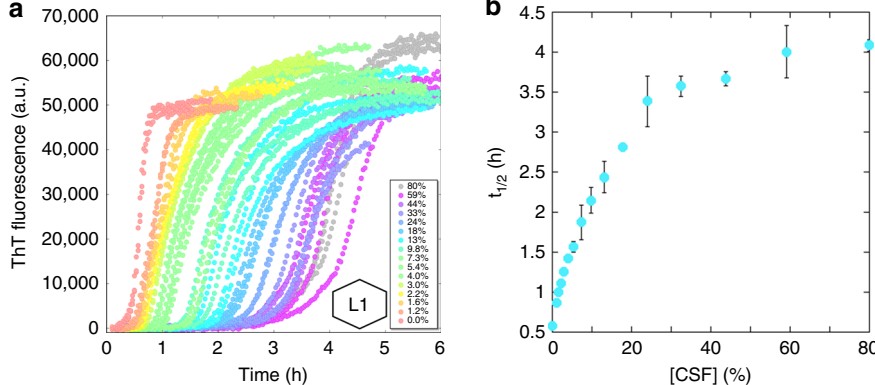

**Fig. 2 Aggregation kinetics in constant Aβ42 concentration. a** The ThT fluorescence as a function of time for 3 μM Aβ42 in buffer with various concentrations of CSF (0–80%). The hexagonal figure represents the CSF pool used. **b** The $t_{1/2}$ as estimated from the data in panel (**a**) as a function of CSF concentration. The error bars are the standard deviations from the $n = 3$ repeats per CSF concentration. The fluorescence data underlying each time trace as well as the $t_{1/2}$ values and the associated standard deviations can be found in Supplementary Data 2

at similar ionic strength and pH (4 mM sodium phosphate, 0.04 mM EDTA, 150 mM NaCl, pH 8.0[33]).

**Effect of CSF concentration.** Next, the overall effect of CSF on Aβ42 aggregation kinetics was investigated at constant Aβ42 concentration as a function of CSF concentration. As several pools of CSF were used, all datasets with CSF have been marked with a hexagonal symbol representing the pool used in the experiment, where L is a pool from Lund and G is a pool from Gothenburg. Solutions of freshly isolated monomeric recombinant Aβ42 were complemented with buffer, salts, ThT, and CSF to obtain a series of samples with varying CSF concentration (0–80%) at constant Aβ42 concentration (3 or 5 μM), and constant concentration of all other components: 10 μM ThT, 20 mM HEPES/NaOH, 140 mM NaCl, 1 mM CaCl₂, pH 8.0. Each experiment was repeated twice with three replicates of each solution per repeat. Examples of data with 3 μM Aβ42 are shown in Fig. 2a and the rest of the data in Supplementary Fig. 3. We find an increase in ThT fluorescence with increasing time and Aβ42 concentrations, comparable to the observations in buffer, strongly indicating the formation of Aβ42 fibrils in CSF. Compared to buffer data, we find a retarding effect of CSF which is prominent already at low (1%) CSF concentration and levels off at high CSF concentrations. At 3 and 5 μM Aβ42, we note that all aggregation profiles (ThT signal versus time) at low and high CSF concentration have a sigmoidal-like appearance with a lag phase, a growth phase, a final plateau, and a relatively symmetric shape

around the midpoint of the growth phase. At intermediate CSF concentrations (ca. 4–20%), the curves are somewhat asymmetric and the initiation of the growth phase is more steep than its end (Supplementary Figs. 3 and 4).

The $t_{1/2}$ value was extracted from each curve for 3 μM Aβ42 in Fig. 2a and plotted versus CSF concentration in Fig. 2b. We note that $t_{1/2}$ versus CSF concentration follows a hyperbolic trend, suggesting that the retarding effect is due to one or more components in CSF interacting with some Aβ42 species along the aggregation pathway. This CSF concentration-dependent retardation was found for two other pools of CSF, one yielding a similar hyperbolic trend (Supplementary Fig. 3b, c)[47,48], while for the other we observed a biphasic effect of $t_{1/2}$ (Supplementary Fig. 3d, e).

This was used to guide the choice of CSF concentrations for a further detailed mechanistic study, to represent one concentration where half maximal effect on $t_{1/2}$ was observed (15%), one concentration where ca. 80% of the maximum effect is observed (32%), and one concentration close to full effect (66%).

The maximum ThT fluorescence intensity between initial baseline and final plateau was evaluated at 3 and 5 μM Aβ42 and shows a small and non-systematic dependence on the CSF concentration (Supplementary Figs. 3–5). The exact pattern deviates between the different trials, as well as the maximum intensity, which could be due to three different pools of CSF being used in these experiments. Since nearly all Aβ42 is consumed at the end of the reaction in the absence of CSF, any increase in ThT intensity must reflect a change in the ThT quantum yield or its affinity to fibrils rather than an increase in amount of fibrils formed.

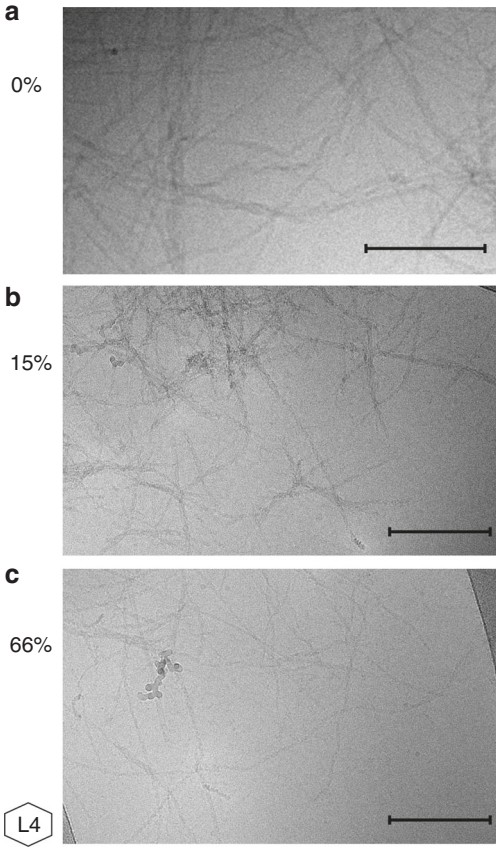

**Fig. 3** Cryo-EM images. Cryo-EM images of the final aggregates formed in solutions with a total monomer concentration of 10 µM and **a** 0%, **b** 15%, or **c** 66% CSF. The hexagonal figure represents the CSF pool used. Scale bar = 200 nm

**Cryo-EM imaging.** To further investigate the formation of fibrils in the presence of CSF, samples of 10 µM Aβ42 in the presence of 15% and 66% CSF, in 20 mM HEPES/NaOH, 140 mM NaCl, 1 mM CaCl$_2$, 10 µM ThT were monitored using the ThT fluorescence assay, collected after reaching the plateau, and imaged using cryo electron microscopy (cryo-EM; Fig. 3b, c). These images show that fibrils are clearly formed also in the presence of CSF, appearing similar to fibrils formed at the same ionic strength without CSF (Fig. 3a)[33], and that they display the hallmark morphological features of amyloid fibrils, being unbranched and ribbon-like with a repeating twist. The presence of CSF does not induce any morphological differences that are discernable with the current method.

**Aβ42 aggregation kinetics at constant CSF concentration.** When comparing different pools of CSF we noted a small discrepancy in fibrillation times depending on the pool used. Therefore, when we next studied the aggregation kinetics starting from a range of Aβ42 monomer concentrations (0.8–10 µM) at constant concentration of CSF (15%, 32%, or 66%), two different pools of CSF were used in a comparative fashion. The pools, originating from two different clinics, were characterized in terms of total protein concentration, ionic strength, and pH (Supplementary Table 1) and used in separate experiments. The concentrations of CSF were chosen as described previously. All samples contained 10 µM ThT, 20 mM HEPES/NaOH, 140 mM NaCl, 1 mM CaCl$_2$, pH 8.0. The experiment was repeated twice at each CSF concentration with either triplicates or quadruplicates of each Aβ42 concentration, meaning in total 6–8 kinetic curves

for each combination of Aβ42 and CSF concentration. The raw data from one repeat at each CSF concentration—with a CSF pool originating from Sahlgrenska Academy, Gothenburg University—are shown in Fig. 4, and from the other repeats—with CSF from Lund University—in Supplementary Figs. 6–8. The data in Fig. 4d, e and Fig. 2a are color coded in the same way according to CSF concentration for easy comparison. Clearly, in all cases the aggregation at each monomer concentration is slower in the presence of CSF compared to in pure buffer.

The half-time, $t_{1/2}$, was extracted from each curve, plotted versus Aβ42 concentration (Fig. 4d, e), and a power function was fitted to the resulting values (Eq. (1)). We find that the scaling exponent is $\gamma = -0.8$ in 15% CSF; $\gamma = -0.9$ in both 32% and 66% CSF, although we note a deviation from the power function, especially at 15% and 32% CSF. All of these $\gamma$ values are lower (more negative) than for Aβ42 in the HEPES buffer with physiological salt, but similar to phosphate buffer at the same ionic strength[33] and higher (less negative) than the value found at moderate ionic strength ($I = 33$ mM, $\gamma = -1.3$)[32,33].

**Kinetic analysis.** To further study the aggregation mechanism of Aβ42—specifically, whether the double nucleation mechanism discovered in pure buffer (Fig. 5) exists also in CSF—a range of kinetic models were globally fitted to each data set using the Amylofit interface[45] to find the minimal model that can reproduce all data. As shown in the Fig. 6a, c, e, g, a model that includes only primary nucleation and elongation cannot fit any of the data obtained in the presence of 0%, 15%, 32%, or 66% CSF, requiring the presence of an additional step, such as secondary nucleation, in all cases.

First, we analyzed the aggregation kinetics data as a function of Aβ42 monomer concentration in 20 mM HEPES/NaOH, 140 mM NaCl, 1 mM CaCl$_2$, pH 8.0. The low dependence on monomer concentration indicates that the reaction order of the secondary process is very low. We therefore chose a model which includes primary nucleation, elongation, and a secondary nucleation process that is explicitly treated as a multi-step mechanism and can thus saturate and lose its monomer concentration-dependence at high monomer concentrations[36]. This model was then used to successfully fit all other CSF concentrations as well; the parameters extracted from this model can be found in Supplementary Table 2 (misfits to an alternative model can be found in Supplementary Fig. 9).

In all of 15%, 32%, and 66% CSF (Fig. 6b, d, h), the aggregation is retarded relative to buffer over the entire monomer concentration range. The delay is reflected in an increase in lag time and all curves have very steep transitions. For the data at 15% CSF, good fits can again be achieved with the same model as in buffer, a mechanism including a multi-step secondary nucleation process. In 15% CSF the saturation of secondary nucleation occurs over the range of concentrations sampled, as indicated by the positive curvature in the half time plots. The aggregation of Aβ42 in high ionic strength phosphate buffer also follows this mechanism[33]. In 32% and 66% CSF the data again display a low monomer dependence as for the buffer case. The fits reflect this; the saturation concentration for secondary nucleation lies well below any of the sampled concentrations, meaning this process is saturated at all concentrations studied. For all three CSF concentrations, the model including primary nucleation, elongation, and fragmentation also gave improved fits compared to models including only primary nucleation and elongation; however, when investigating the mean residual error (MRE) as calculated by Amylofit, we found that a model including fragmentation instead of multi-step secondary nucleation had a higher MRE for all CSF concentrations, namely 165% increase for

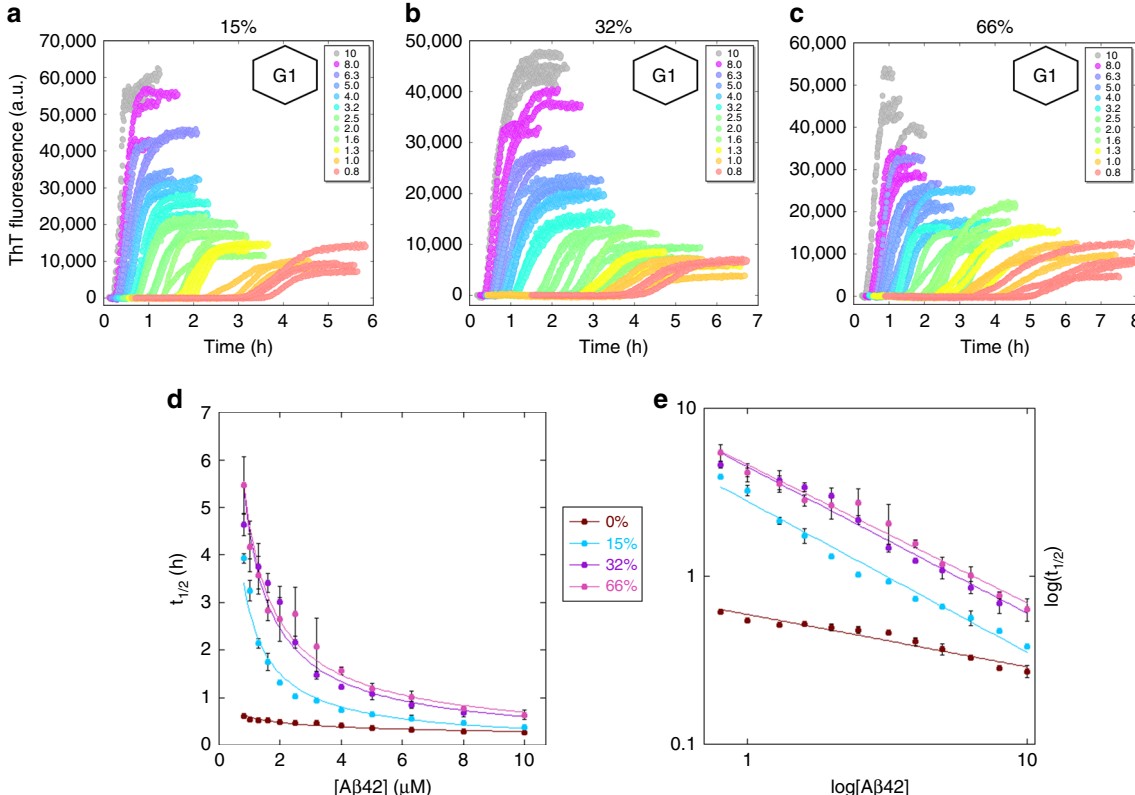

**Fig. 4** Aggregation kinetics in CSF. The ThT fluorescence as a function of time (h) for 0.8–10 μM Aβ42 in buffer with **a** 15% CSF; **b** 32% CSF; **c** 66% CSF. The hexagonal figures represent the CSF pool used. **d** is the calculated $t_{1/2}$ (h) as a function of Aβ42 concentrations for all CSF concentrations; it is also plotted with a fitted power function, Eq. (1). **e** is the double log representation of (**d**). The error bars are the standard deviations from the $n = 4$ repeats per Aβ42 concentration. The fluorescence data underlying each time trace as well as the $t_{1/2}$ values and the associated standard deviations can be found in Supplementary Data 3

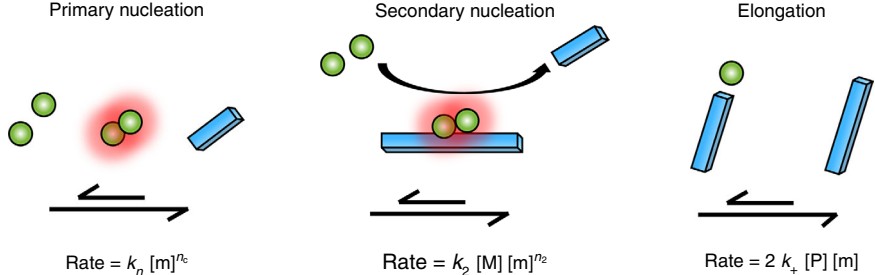

**Fig. 5** The double nucleation mechanism. The double nucleation mechanism discovered for Aβ42 in a phosphate buffer[32]. Primary nucleation is a reaction starting from monomers in solution and in the kinetic description used here leads directly to growth competent species. The rate at which this process yields new fibrils, [P], is given by $k_n[m]^{nc}$, where [m] is the monomer concentration, [P] is the fibril concentration in fibril units and $n_c$ is the reaction order, which constitutes a lower bound for the number of monomers in the nucleus. Secondary nucleation is a reaction whereby monomers from solution react on the fibril surface and in the kinetic description used here leads directly to growth competent species in solution. The rate at which this process yields new fibrils, [P], is given by $k_2[M][m]^{n2}$, where [m] is the monomer concentration, [M] is the fibril concentration in monomer units, and $n_2$ is the reaction order, which constitutes a lower bound for the number of monomers in the nucleus. Elongation is a reaction whereby monomers add to the fibril ends to extend the fibril by one monomer unit. The rate at which this process yields new fibril mass, [M], is given by $2k_+[P][m]$, where [m] is the monomer concentration

15% CSF over 24,188 data points; 3.2% increase for 32% CSF over 36,988 data points; and 6.7% increase for 66% CSF over 36,096 data points (Supplementary Fig. 10). Given this observation, along with the identification of multi-step secondary nucleation as the main mechanism in phosphate buffer at the same ionic strength[33], we conclude multi-step secondary nucleation to be the more likely mechanism in CSF as well.

The key finding of the kinetic analysis, which is consistent across all CSF concentrations, is thus a failure to fit the data unless a secondary nucleation pathway is included in the model.

Indeed, models that lack this pathway fail to reproduce the data (Fig. 6a, c, e, f). This implies that the secondary nucleation pathway, discovered for Aβ42 in pure buffer system[32], is a dominant microscopic mechanism for the generation of new aggregates also in CSF.

**Seeding experiments**. To further verify the presence of a secondary pathway, the effect of preformed fibrils on aggregation time was investigated (Fig. 7). Seed fibrils were prepared from 3

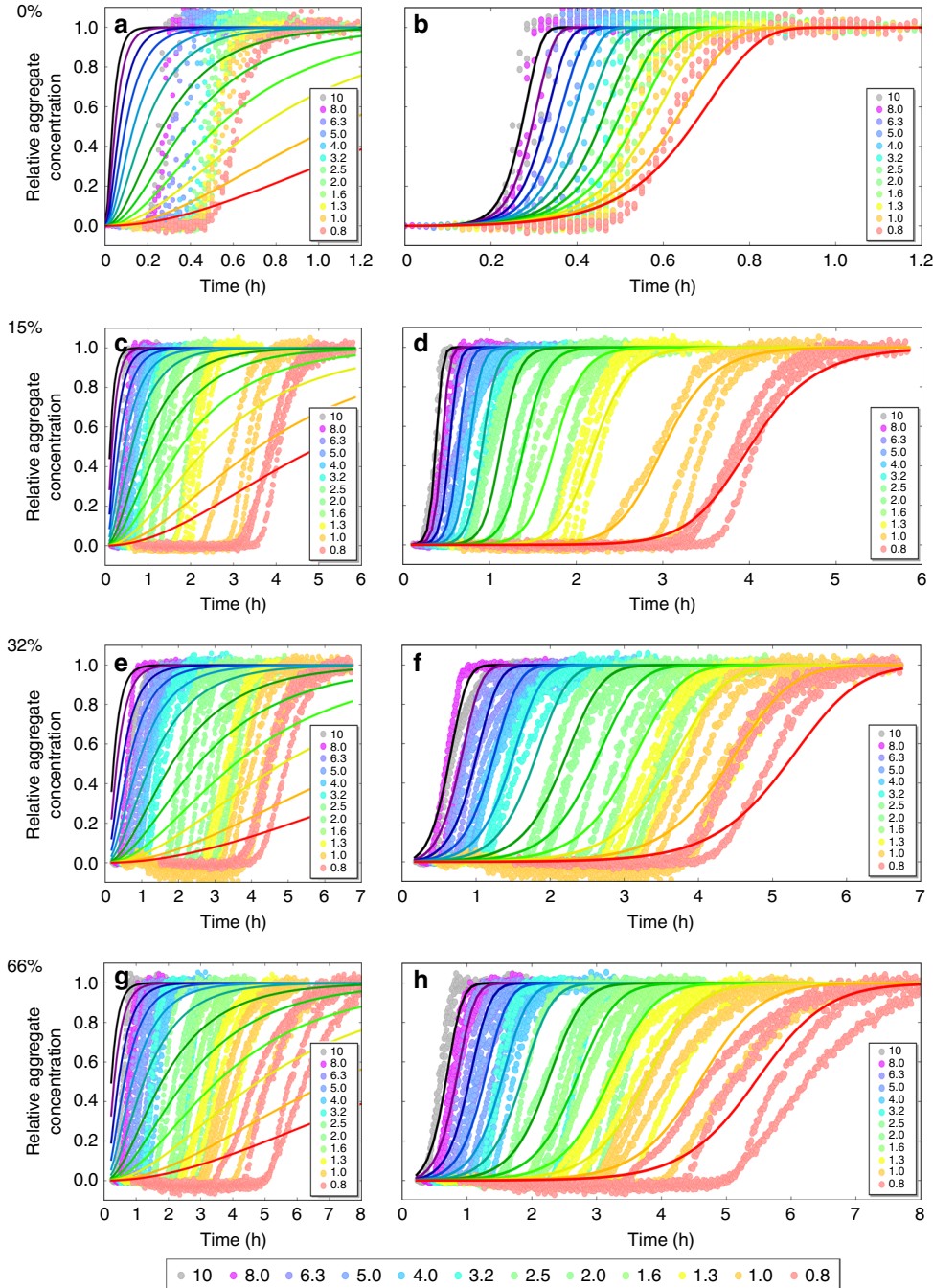

**Fig. 6** Normalized kinetics with fitted models. Normalized ThT fluorescence (relative aggregate concentration) as a function of time (h) for 0.8–10 μM Aβ42 in buffer with **a, b** 0% CSF, **c, d** 15% CSF, **e, f** 32% CSF, and **g, h** 66% CSF, together with fits obtained using AmyloFit[45]. Graphs (**b, d, f, h**) show the fit using a multi-step secondary nucleation dominated mechanism, while (**a, c, e, g**) show the same data fitted with a model including primary nucleation and elongation only. The legend at the bottom shows the color coding of the concentrations of Aβ42 in μM. The normalized fluorescence data as well as the two fits evaluated at each time point can be found in Supplementary Data 4

μM monomeric Aβ42 solutions in 20 mM HEPES/NaOH, 140 mM NaCl, 1 mM CaCl₂, 10 μM ThT, pH 8.0 with 0% and 66% CSF, respectively. The seed fibrils were then added to 3 μM monomeric Aβ42 solutions in the same solution conditions to final seed concentrations of 30%, 10%, 3%, 1%, and 0% of the monomer concentration in monomer equivalents. In both conditions the addition of seeds shortens the lag phase in a non-linear fashion and even the lowest seed concentration (1%) leads to a considerable shortening of the lag phase, as shown also in the $t_{1/2}$-plots. Even at the highest seed concentration the aggregation

does not start immediately creating an apparent lag phase, which may be partly due to the time it takes for the solution to reach 37 °C.

We attempted to make fits to the seeded data using models and rate constants that were consistent with the results from the unseeded experiments (Supplementary Fig. 11). Again it was clear that a model including only nucleation and elongation could not reproduce the data, the inclusion of secondary processes is necessary, especially to capture the strong effects of low seed concentrations. Only the model relying on secondary nucleation

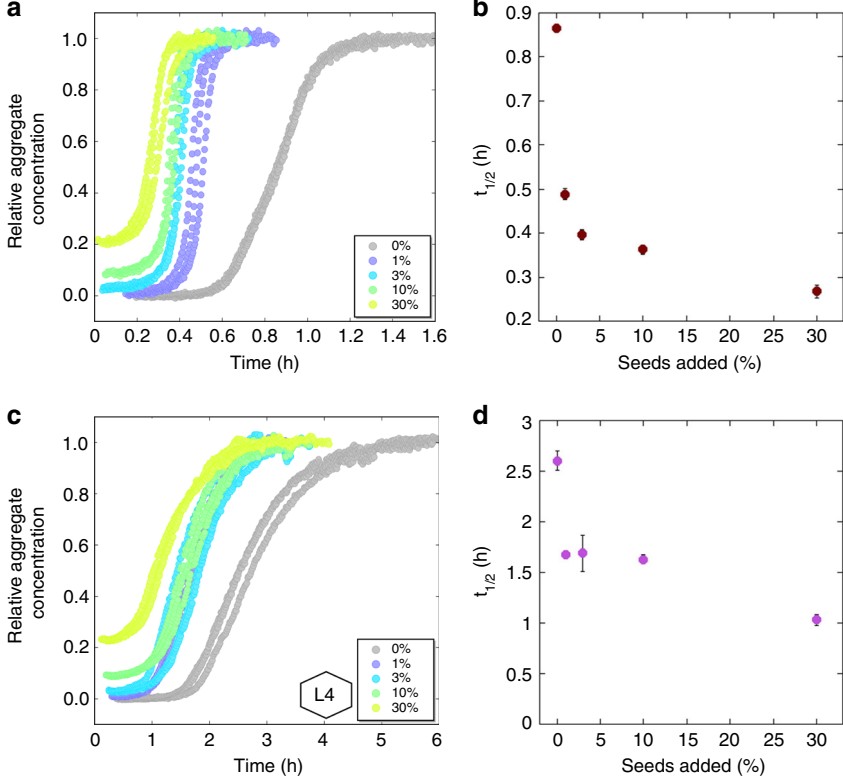

**Fig. 7 Seeded data.** Normalized fluorescence intensity from seeded aggregation experiments with 3 µM Aβ42 monomer concentration in **a** buffer and **c** 66% CSF and addition of 0–30% seeds. The hexagonal figure represents the CSF pool used. **b** and **d** are the estimated values of $t_{1/2}$ (h) as a function of concentration of the added seed in %. The error bars are the standard deviations from the $n = 3$ repeats per seed concentration. The fluorescence data underlying each time trace as well as the $t_{1/2}$ values and the associated standard deviations can be found in Supplementary Data 5

gave fits that could reproduce both the shape of the curves and the dependence on seed concentration. However, the fits to the seeded data were found to be significantly worse than fits to the unseeded data, which may be due to components from CSF interacting with mature fibrils and altering their seeding abilities.

## Discussion
The mechanism of Aβ aggregation in vivo remains elusive, although kinetic studies in pure buffer provide clues as to which microscopic steps might be present. In this study, we have expanded these studies towards an in vivo-like situation using human CSF spiked with recombinant human Aβ42 peptide. Based on the results we can conclude the following:

Aβ42 forms fibrils in CSF. Although the presence of CSF certainly delays fibril formation[49–51], Aβ42 is found to form ThT-positive aggregates at all CSF concentrations studied (up to 90%). The cryo-EM images clearly show that fibrils are formed in samples containing 15% or 66% CSF, with a morphology resembling those formed in buffer at similar ionic strength[33]. Curiously, in several fields of the images, small particles seem to be associated with the surface of the fibrils and these features are more prevalent in images taken at the higher concentration of CSF. This is interesting in light of the identification of HDL-like components in the retarding fraction of size-separated CSF[49]. However, the cryo-EM images alone do not allow for identification of the bound species and further investigation would be required.

Reproducible data amenable to kinetic analysis can be obtained in CSF. Kinetic analysis requires high quality, reproducible data. The addition of a complex biological fluid such as CSF could potentially interfere with the aggregation kinetics in a number of ways, and we ask what level of reproducibility can be achieved between and within CSF pools. Most notably, any body fluid will contain thousands of high and low Mw proteins or other components with varying influence on the aggregation process. The kinetic data will report on a *net effect* of all these components and a variation in the composition of the CSF may modulate this net effect. Five CSF pools from healthy donors collected at two different clinics were used in the current study. The data in Figs. 2, 3, 7, and Supplementary Figs. 2–9 were obtained using four CSF pools from Lund University Hospital, whereas data obtained using one CSF pool from Sahlgrenska Academy, Gothenburg University are shown in Figs. 1, 4, and 6. Although some minor differences in curve shape can be detected when comparing the data at 66% CSF from Gothenburg with the data at 66% CSF from Lund, the overall appearance is very similar and the observed $t_{1/2}$ versus Aβ42 concentrations superimpose closely (Fig. 4c, d, Supplementary Fig. 8). The slight variation between kinetic curves, as well as the variation seen between pools in terms of CSF concentration dependence at fixed Aβ42 concentration, is most likely due to variations between the individuals donating the CSF (a comparison of some general properties between two pools of CSF is summarized in Supplementary Table 1). Still, the reproducibility between replicates of the same Aβ42 concentration found at 15% as well as 32% and 66% CSF is remarkably high. Some discrepancy between the $t_{1/2}$ times from Fig. 2 and Figs. 1 and 4 can be noted, which could due to different CSF pools being used; however, the macroscopic behavior—with retardation at higher CSF concentration as well as saturation of the effect—remains the same over all CSF pools used. In addition, there are potential practical complications associated with the use of biological fluids, including difficulties of accurately dispensing a viscous fluid, loss of fluorescent signal due to interfering

chromophores or spurious signals from unknown aggregating proteins in solution. The first two issue do not pertain to CSF which is a transparent, Newtonian fluid with a viscosity close to that of water[52]. We cannot rule out that CSF itself might contain proteins that aggregate and bind ThT, or co-aggregate with the added Aβ42, as we do see a slight variation in the amplitude of the fluorescence change in some of the CSF pools. However, when CSF was incubated with ThT without added Aβ42, some increase of ThT fluorescence was observed within 25 h (Supplementary Fig. 3a), but very much lower than the change in the presence of Aβ42; therefore, this does not interfere with reproducibility, nor hamper our analysis.

The models of aggregation of pure protein are able to reproduce the data in pure buffer (Fig. 6a). Although they perform less well in the presence of CSF, with some deviations in curve shape, the overall behavior and concentration dependence is fitted surprisingly well. This observation indicates that the presence of the wide range of biological compounds contained in CSF can be approximated simply as perturbations to the rate constants of aggregation, leaving the basic mechanism unchanged. The deviation in curve shape could possibly be caused by interactions in CSF between Aβ42 and some species, with the dissociation of any formed complexes being rate limiting and affecting the reaction profile; however, further studies would be necessary for full disclosure of the possibilities behind the deviations. Still, the present findings show that it is feasible to study the aggregation kinetics of Aβ peptides in CSF, with high reproducibility between replicates of considerable value for drug discovery programs. Indeed, a recent study[47] used this opportunity to validate that small molecules, discovered to retard specific microscopic steps in the Aβ42 aggregation process in buffer, caused prominent retardation also in CSF as a representative of a more natural environment in addition to a pure buffer system.

The autocatalytic nature of the process, including a double nucleation mechanism originally found in pure buffer, is prevalent in CSF. In HEPES buffer with physiological salt (20 mM HEPES/NaOH, 140 mM NaCl, 1 mM CaCl$_2$, pH 8.0), good global fits to the data could be achieved by a double nucleation model including primary nucleation of monomers in solution and multi-step secondary nucleation of monomers on fibril surface (Fig. 6b). The same mechanism was identified in a recent study in phosphate buffer at the same ionic strength (4 mM sodium phosphate, 0.04 mM EDTA, 150 mM NaCl, pH 8.0[33]) with the main difference being that both primary and secondary nucleation seemed to follow a reaction order of two in the latter case. The effects of adding premade fibrils to the reaction mixture, i.e. shortening of the lag phase in a manner that can only be explained by the presence of secondary processes, further supporting this.

The aggregation mechanism can be described in the same manner as in pure buffer, but the rate constants for the individual microscopic steps are affected by CSF. Having found a unifying model that can describe all the data in buffer and at both CSF concentrations tested, it is interesting to analyze how the parameters differ between the conditions. The most consistent effect of adding CSF is a decrease of the secondary nucleation rate. This reduction might be connected to the small objects observed along some fibrils in the cryo-EM images that could possibly be a component of CSF binding to the surface of the fibrils and thus reducing the rate of secondary nucleation on this surface. Future work might examine if there are differences in Aβ aggregation characteristics in CSF obtained from individuals with and without biomarker evidence of cerebral β-amyloidosis.

To summarize, we found that fibrils are formed in the presence of CSF and are of similar morphology as those formed in buffer. Monitoring of ThT fluorescence yielded reproducible kinetic data of aggregation in CSF, which could be analyzed by global fitting of

the same rate laws that apply to the aggregation in pure buffer. This highlights that the presence of CSF can be approximated as a perturbation to the rate constants of aggregation in pure buffer and, most importantly, that the autocatalytic nature of the process, including a double nucleation mechanism, is present also in CSF. The main retarding effect of CSF on the aggregation of Aβ42 stems from a reduction of the rate of secondary nucleation. The decrease in secondary nucleation rate suggests that an inhibitor of secondary nucleation may be responsible for the retardation.

Our findings illustrate how aggregation kinetics in pure buffer can identify the microscopic steps in the aggregation reaction and then be used to translate these findings to the in vivo situation by repeating the aggregation reaction in increasing concentrations of bodily fluids. By identifying which processes are most affected this approach points towards possible mechanisms of action and can quantify the effect on different steps and under different conditions. The ability to study aggregation at a quantitative, mechanistic level in bodily fluids is key for investigating the differences in CSF obtained from individuals with and without AD and may help determine the presence of disease enhancing or retarding factors in different individuals.

## Methods

**Validation of automated pipetting by robot**. All dilutions of Aβ monomers in this work were performed by an in-house constructed automated dispenser robot. The accuracy and precision of this robot was validated by creating dilution series of a fluorescent dye. Buffer A was created by dissolving pyranine dye (8-hydroxypyrene-1,3,6-trysulfonic acid trisodium salt, Invitrogen) to a concentration of 2 μM in 50 mM bicine–HCl, pH 9.0. Buffer B consisted of pure 50 mM bicine–HCl, pH 9.0. The pipetting robot was used to dispense 24 different combinations of the two buffers, ranging from 100 μL A + 0 μL B to 0 μL A + 100 μL B into separate wells in a 96-well plate. Each set contained four replicates of every concentration and three independent sets were made. The fluorescence intensities of the resulting dye solutions were recorded at 510 nm with an excitation light at 450 nm in a CLARIOstar plate reader (BMG Labtech).

**CSF pools**. De-identified pooled CSF from healthy donors was used in all experiments. Five different pools were used: one from the Clinical Neurochemistry Laboratory, Sahlgrenska University Hospital, Gothenburg, and four from Lund University Hospital. Each pool was frozen at −80 °C as multiple identical 1 mL aliquots. Upon thawing, the CSF sample was supplemented with 20 mM HEPES from a 30× concentrated stock to adjust pH to 8.0. Assuming literature values of salt concentrations, the CSF stock was then assumed to contain 97% CSF, 20 mM HEPES/NaOH, 140 mM NaCl, 1 mM CaCl$_2$, pH 8.0. The CSF pool was prepared from de-identified left-over aliquots from clinical routine analyses, following a procedure approved by the Ethics Committee at University of Gothenburg (EPN 140811).

**Characterization of CSF pools**. A comparison between one pool of CSF from the Clinical Neurochemistry Laboratory, Sahlgrenska University Hospital and one pool from Lund University Hospital was made, in terms of pH, conductivity (ionic strength), and overall protein content. The pH was measured using a Mettler Toledo MP225 pH-meter; the conductivity was measured using a CONSORT C830 pH/mV/Conductivity/°C-meter; finally, the total protein content was determined using the Bradford assay[53], with human IgG as standard.

**Cloning and expression of Aβ peptides**. A synthetic gene with E. coli optimized codons for Aβ(M1-42) was produced by PCR from overlapping oligonucleotides and cloned into PetSac, a derivative of Pet3a plasmid[54]. The peptides were expressed in E. coli strain BL21 DE3 PLyS star (Invitrogen) in LB medium with 50 mg L$^{-1}$ ampicillin and 30 mg L$^{-1}$ chloramphenicol. Well-isolated small colonies from bacteria freshly transformed with vector were used to inoculate 50 mL overnight cultures grown in 250 mL baffled flasks at 37 °C with 125 rpm shaking. These were diluted 1:100 into 500 mL day cultures grown in 2500 mL baffled flasks at 37 °C with 125 rpm shaking. 0.4 mM IPTG was added at OD 0.6–1.0 and the cells were harvested after another 3.5–4 h by centrifugation for 10 min at 6000 × g.

**Purification of Aβ peptides**. Cell pellet from 2 L culture was sonicated in 40 mL 10 mM Tris/HCl, 1 mM EDTA, pH 8.5 (buffer A) using a sonicator tip (half horne, 50% duty cycle, maximum output, 30–90 s). This step was performed in a glass beaker surrounded by an ice–water slurry. Inclusion bodies were isolated by centrifugation at 4 °C, 15,000 × g for 10 min. Two more rounds of sonication and

centrifugation were performed. The inclusion bodies were dissolved in 40 mL 8 M urea in buffer A, and diluted four-fold in buffer A. 20 mL DEAE cellulose (Whatman DE23, equilibrated in buffer A with 2 M urea) was added and the slurry was left on ice for 30 min (with stirring now and then using a glass rod). The solution was removed and the resin was washed with buffer A in a Büchner funnel on a vacuum flask, followed by two washes with 30 mL buffer A with 25 mM NaCl and elution in buffer A with 50 mM NaCl, 6× 30 mL or 125 mM NaCl 3× 30 mL. The ion-exchange purification was performed in batch mode to avoid concentration of the peptide on the resin. The eluted fractions were examined using agarose gel electrophoresis and SDS–PAGE. Fractions containing Aβ42 were pooled, lyophylized, dissolved in 6 M GuHCL and again purified by SEC on a 2.6 × 60 cm Superdex 75 column in 20 mM NaP, 0.2 mM EDTA, pH 8.5. The eluted fractions were examined using UV absorbance and SDS–PAGE. Pure fractions were pooled, aliquoted, lyophilized, and stored at −20 °C.

**Chemicals**. All chemicals were of analytical grade. Buffers were extensively degassed.

**Preparation of samples for kinetics experiments**. Aliquots of purified Aβ42 were dissolved in 6 M GuHCl, and monomer isolated by gel filtration on a Superdex 75 column in 20 mM HEPES/NaOH, pH 8.0. The gel filtration steps remove traces of pre-existent aggregates and exchanges the buffer to the one used in the fibril formation experiments. The peptide concentration was determined from the absorbance of the integrated peak area using $\varepsilon_{280} = 1400\ \mathrm{L\ mol^{-1}cm^{-1}}$ as calibrated using quantitative amino acid analysis. The concentration determined by this method is correct within ± 20% or better. The monomer generated in this way was diluted with buffer to 10 μM and supplemented with 10 μM thioflavinT (ThT, Calbiochem) from a 2 mM stock (in water, filtrated through 0.2 μm filter). All samples were prepared in low-binding Eppendorf tubes (Axygen, California, USA) on ice using careful pipetting to avoid introduction of air bubbles. The monomer sample was supplemented with 140 mM NaCl and 1 mM $CaCl_2$. The CSF concentration series was prepared from a monomer stock, buffer, and CSF to yield samples with none or 1.2–80% CSF. The Aβ42 concentration series were prepared from samples with CSF concentrations of either 0%, 15%, 32%, or 66% along with buffer, and Aβ42 to yield samples with 10–0.8 μM Aβ42. Each sample was pipetted into three or four wells of a 96-well half-area plate of black polystyrene with a clear bottom and PEG coating (Corning 3881, Massachusetts, USA), 100 μL per well, using the pipetting robot.

**Kinetic assays**. Assays were initiated by placing the 96-well plate at 37 °C under quiescent conditions in a plate reader (Fluostar Omega or Fluostar Optima BMGLabtech, Offenburg, Germany). The ThT fluorescence was measured through the bottom of the plate every 60 s with a 440 nm excitation filter and a 480 nm emission filter.

**Seeded assays**. The seed fibrils were prepared from 3 μM monomer solutions (in 20 mM HEPES/NaOH, 140 mM NaCl, 1 mM $CaCl_2$, pH 8.0, 10 μM ThT, as above) supplemented with 0% or 66% CSF and incubated at 37 °C under quiescent conditions in a plate reader until the ThT fluorescence reached a plateau value. The seed fibrils were added to monomer solutions (in 20 mM HEPES/NaOH, 140 mM NaCl, 1 mM $CaCl_2$, pH 8.0, 10 μM ThT, as above). The final seed concentrations were 30%, 10%, 3%, 1%, and 0% of the monomer concentration in monomer equivalents. Each mixture was then pipetted into three different wells in a 96-well plate.

**Cryo-EM**. Samples containing 10 μM Aβ42 in either 15% or 66% CSF, 10 μM ThT, 20 mM HEPES/NaOH, 140 mM NaCl, 1 mM $CaCl_2$, pH 8.0 were prepared using the ThT fluorescence assay as described above. After reaching the plateau, the samples were put on ice. Specimens for electron microscopy were prepared in a controlled environment vitrification system (CEVS) to ensure stable temperature and to avoid loss of solution during sample preparation. The specimens were prepared as thin liquid films, <300 nm thick, on lacey carbon-filmed copper grids and plunged into liquid ethane at −180 °C. This leads to vitrified specimens, avoiding component segmentation and rearrangement, and the formation of water crystals, thereby preserving original microstructures. The vitrified specimens were stored under liquid nitrogen until measured. A Fischione Model 2550 cryo transfer tomography holder was used to transfer the specimen into the electron microscope, JEM 2200FS, equipped with an in-column energy filter (Omega filter), which allows zero-loss imaging. The acceleration voltage was 200 kV and zero-loss images were recorded digitally with a TVIPS F416 camera using SerialEM under low dose conditions with a 30 eV energy selecting slit in place.

**Kinetic analysis**. Determination of the scaling factor was done by fitting the described equation to the $t_{1/2}$ plots. This was done by using the power function in KaleidaGraph, ver. 4.1.1.

Global analysis of Aβ42 aggregation kinetics to extract the rate constants for primary nucleation, secondary nucleation, and elongation was performed using the

online Amylofit platform[45]. At each condition (buffer only, buffer with 15% CSF, buffer with 32% CSF, or buffer with 66% CSF) fits were made to the data using the following integrated rate law for the normalized aggregate concentration

$$\frac{[M]}{[M]_\infty} = 1 - \left(1 - \frac{[M]_0}{[M]_\infty}\right)e^{-k_\infty t} * \left(\frac{B_- + C_+ e^{\kappa t}}{B_+ + C_+ e^{\kappa t}} * \frac{B_+ + C_+}{B_- + C_+}\right)^{\frac{k_\infty^2}{k_\infty \kappa}} \qquad (1)$$

where the parameters are defined as follows

$$\kappa = \sqrt{2[m]_0 k_+ \frac{[m]_0^{n_2} k_2}{1 + [m]_0^{n_2}/K_M}} \qquad (2)$$

$$\lambda = \sqrt{2k_+ k_n [m]_0^{n_c}} \qquad (3)$$

$$C_\pm = \frac{k_+[P]_0}{\kappa} \pm \frac{k_+[M]_0}{2[m]_0 k_+} \pm \frac{\lambda^2}{2\kappa^2} \qquad (4)$$

$$k_\infty = 2k_+[P]_\infty \qquad (5)$$

$$\bar{k}_\infty = \sqrt{k_\infty^2 - 2C_+ C_- \kappa^2} \qquad (6)$$

$$B_\pm = \frac{k_\infty \pm \bar{k}_\infty}{2\kappa} \qquad (7)$$

In these relations, $[m]_0$ is the initial monomer concentration, $[P]_0$ is aggregate number at the start of the reaction, $[P]_\infty$ is the aggregate number at equilibrium, that is, after reaction completion, $[M]_0$ is the mass concentration of fibrils at the start of the reaction and $[M]_\infty$ is the mass concentration of fibrils at equilibrium. $k_n$, $k_2$, $k_+$ are the rate constants for primary nucleation, secondary nucleation, and elongation, respectively. $K_M$ is the saturation constant for secondary nucleation and $n_c$ and $n_2$ are the monomer scalings of primary and secondary nucleations, respectively. The values $n_c = 0.00001$, and $n_2 = 2$ were used for fitting the kinetics in buffer, and the values $n_c = 3$ and $n_2 = 2$ for the fitting of all aggregation kinetics in the presence of CSF.

The effective rate of formation of new aggregates through secondary nucleation at a given concentration was calculated using the following relation:

$$\frac{dP_2}{dt} = k_2[M]\frac{m^{n_2}}{1 + \frac{m^{n_2}}{K_M}} \qquad (8)$$

**Statistics and reproducibility**. In this work each experiment consists of a dilution series created by the pipetting robot by dispensing different volumes of stock solution containing Aβ42 and/or CSF and buffer into a 96-well plate to achieve the desired concentrations. Replicates within one experiment are created by dispensing the same volumes in several consecutive wells. Independent repeats of experiments were made always made using new stock solutions, when new CSF pools were used this has been indicated. All error bars are showing the standard deviation of such replicates within one experiment. Separate repeats of the same experiments are always shown in separate figures or panels.

**Reporting summary**. Further information on research design is available in the Nature Research Reporting Summary linked to this article.

## Data availability

Raw data are shown in Figs. 1–4 and Supplementary Figs. 1–4, 6–8. Normalized data are shown in Figs. 6, 7 and Supplementary Figs. 5, 9, and 11. The data underlying Fig. 1 are shown in Supplementary Data 1. The data underlying Fig. 2 are shown in Supplementary Data 2. The data underlying Fig. 4 are shown in Supplementary Data 3. The data underlying Fig. 6 are shown in Supplementary Data 4. The data underlying Fig. 7 are shown in Supplementary Data 5. All other data will be made available upon request.

## Code availability

The software used to analyze the data can be accessed at http://www.amylofit.ch.cam.ac.uk.

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

## Acknowledgements

The expert help with cryo-EM by Anna Carnerup. Lund University, is gratefully acknowledged. This work was funded by the European Research Council (S.L.), The Swedish Research Council (S.L.), The Knut and Alice Wallenberg Foundation (S.L.) and Alzheimerfonden (S.L., O.H., H.Z., K.B.). H.Z. is a Wallenberg Academy Fellow. Open access funding provided by Lund University.

## Author contributions

S.L., T.C., K.B., H.Z., and U.A. designed the study; O.H., K.B., H.Z., U.A., B.F. and T.L. contributed with new reagents and tools; R.F., M.T., and S.L. performed the experiments and analyzed the data together with G.M., T.P.J.K. and T.L.; R.F., M.T., and S.L. also wrote the manuscript, with input from all authors.

## Competing interests

The authors declare no competing interests.
