## [Peer Review File · Communications Biology]

Reviewers' comments:

Reviewer #1 (Remarks to the Author):

In the manuscript "Autocatalytic amplification of Alzheimer-associated A β 42 peptide aggregation in human CSF" the authors use their well-established method to quantitatively characterise the microscopic steps in A β 42 aggregation in human CSF. Overall the manuscript are timely, well written and constitute an important step towards understanding peptide aggregation in more complex and relevant solvents. The initial part describes the retardation of fiber formation by CSF and they show that the effect levels off and is saturated at 50-60% CSF. The second part regards identification and quantification of the affected microscopic rate constants, underlying the observed retardation. Although I have some serious concerns on the kinetic analysis, the study breaks new ground and could be of general interest if some clarifications are made:

Major concerns:

(i) My major concern regards the kinetic analysis, where the model used clearly do not apply for the aggregation kinetics in CSF. In buffer the fits are good, and the model seems to catch the overall macroscopic behaviour. However, for both CSF concentration (15% and 66%) the model seems to break down. The fitted functions do not at all fit the experimental data. Nevertheless the authors use the fitted microscopic rates to identify the mechanism for the aggregation retarding effect. The finding that the model is not applicable for A β 42 aggregation in CSF is indeed intriguing and important, but the authors oversees this, and simply uses the fitted values for their analysis. In my opinion, a detailed discussion, and preferably control experiments, on the reason for the model breakdown would strengthen the study significantly, and provide a line of further studies to understand the seemingly fundamental difference between A β 42 aggregation in buffer and CSF. The reproducibility within a CSF batch seem to be high, so the misfit should not be due to ambiguous data, but rather to that the aggregation mechanism alters and is dominated by steps not included in the model. The conclusions from comparing the fitted microscopic rate constants should be omitted, as they do not seem to report on the data.

(ii) Also the seeding experiments show significant systematic deviations from the model. Maybe it would be better to just withdraw model free observables from the data, such as $t_{1/2}$ and maximum growth rate, to quantify the effect. Using the fitted values, again, do not report on the actual data, but merely on the fits. The beauty of the established model is that it captures the macroscopic characteristics and provides the microscopic rates, but in CSF, the model does not capture the macroscopic behaviour, and thus the obtained microscopic rate constants and saturation concentrations are very hard to interpret.

(iii) Third, the differences in A β 42 aggregation in the different CSF batches are interesting and could provide some additional information on the mechanism of the observed retardation. It would be of importance if the authors could characterise and report on the different CSF batches in terms of general properties, such as pH, ionic strength, overall protein concentration and trace metals. This could give a hint on the mechanism(s) that could be further tested.

Minor issue:

Overall, the figures (in the manuscript and supporting information) need to be edited for readability. Both the data legends and axis labels on inserts are too small.

Reviewer #2 (Remarks to the Author):

Frankel et al describes an aggregation kinetics study of Abeta42 in the absence and presence of human CSF. The authors performed several series of ThT kinetics assays in buffer with different Abeta42 monomer concentrations, in solutions containing varying concentrations of human CSF at constant monomer concentration, and in solutions containing 15% or 66.6% CSF with different Abeta42 monomer concentrations. The authors also reported cryo-TEM images of fibrils formed in CSF and seeded experiments using the fibrils generated in corresponding CSF conditions.

I completely agree with the complementary importance of the "top down" approach the authors employ here and I think studying aggregation kinetics in CSF is timely for the advancement of the field, and will help to bridge the gap between in vitro and in vivo knowledge. In fact, the observation by ThT assays and cryo-TEM reported here that fibrils with similar ThT growth traces and morphology can form in the presence of CSF compared with buffer is an important and satisfying result that validates previous research direction taken by the community using pure buffer to resolve the molecular principles involved in Abeta and amyloid aggregation. The data reported in this manuscript is also of extremely high quality with the same high standards as previously published by the authors. The experiments are also very well documented. Therefore, overall I am very happy to see this manuscript, after revision, published so that the results and the data are made available to the community as soon as possible. My major comments, listed below, are mainly on the variability of the data and the model dependent interpretation of the data, and I very much hope the authors will find these useful in revising the manuscript.

- The authors report the use of four different pools of CSF. The authors also carefully documented and reported several different sets of ThT data, presumably repeated using different CSF pools. It is clear for example in supplementary Fig S4 that biological replicates of CSF do result in quite large variation in concentration dependent behaviour. Therefore, I think it will be very useful if the authors can label their datasets with the CSF pool number (e.g. as shown in Fig S4 simply with a number without breaching any ethical issues) so that data from the same pool can be linked through the manuscript.

- The authors presented data with a constant 3uM concentration of Abeta42 and increasing concentrations of CSF. These data indicated an increase in the length of the lag phase as the concentration of CSF is increased. However, in later datasets this trend is apparently not preserved (e.g. the rate of the reactions in 15% CSF is faster than in 66% CSF). Is this a case of biological CSF pool variability? Since one of the conclusions is that the CSF may be giving an saturable inhibitory effect, I think it is important that the authors comment/clarify this. I understand the use of CSF may be very technically challenging, but if possible, I suggest the authors at least one more concentration of CSF to indicate what the actual trend is for the apparent saturation effect observed.

- I understand completely the complexities in model fitting involved in the analysis of ThT data and I agree that the best fit reported by the authors give significantly better results than primary nucleation alone. However, it is still not easy for the readers to decipher the relative qualities of the models as well as other models not visually represented that the authors may have tested. For example, by eye some of the models presented in the supplementary data appear to fit better to the suggested model in the main text. Therefore I think the authors should report a table with some kind of quality of fit measure, assessment criteria, as well as the number of fitted parameters in the text or in the SI. Such inclusion would be extremely useful for the readers.

- In the text page 9, when discussing the saturation of secondary nucleation in pure buffer, the authors refer to figure 4A, which is not pure buffer but 15 % CSF according to the figure legend.

- The top labels for Fig S8 is not consistent with the descriptions in the figure legend.

- It is very interesting that the authors observe small particles that seem to be associated with the fibrils at higher concentrations. The authors suggest that they may be HDL-like components. If these particles are aggregates formed by A β 42 from a competing pathway then that could potentially also contribute to the saturation effect the authors observe. Therefore, I think the discussions of the mechanistic origin of the saturation effect could be expanded in the discussions section to take into such possibilities.

Reviewer #3 (Remarks to the Author):

The paper entitled: "Autocatalytic amplification of Alzheimer-associated A β 42 peptide aggregation in human CSF" is focused in the effect of CSF in amyloid aggregation kinetics. It is vastly known the lag of information about the amyloid aggregation of amyloid proteins in vivo. The main problems are the lag of reproducibility inter-assays and the low protein concentration present in transgenic animals. Thus, currently practically the unique possibility to track the amyloid aggregation in-vivo is in bacteria (in-cellulo assays) tracking the formation of inclusion bodies. In my opinion the paper is nice, interesting and well worked but I have two main constrains.

On one hand, I observe a discrepancy between figure 2 and the consecutive ones. In the figure 2 the authors show an increment of $t_{1/2}$ with CSF concentration (in brief the increment of CSF% increase the $t_{1/2}$). However in the other figures (see e.g. Fig6E), the authors show that 15% CSF inhibit amyloid aggregation more powerfully that 66% CSF using the same Ab42 range. I don't understand this. In addition the kinetics shown in figure 2 and the others display different kinetics (lag time, $t_{1/2}$) at similar Ab42 concentration and CSF%.

On the other hand, although the authors only suggest (propose posterior studies) that effect of CSF is consequence of potential interactions with phospholipids and other compounds presents in CSF, the unique evidence of this in the paper is the presence of several structures shown by TEM. I propose to soften these affirmations in this initial paper.

Reviewers' comments:

Reviewer #1 (Remarks to the Author):

In the manuscript "Autocatalytic amplification of Alzheimer-associated A β 42 peptide aggregation in human CSF" the authors use their well-established method to quantitatively characterise the microscopic steps in A β 42 aggregation in human CSF. Overall the manuscript are timely, well written and constitute an important step towards understanding peptide aggregation in more complex and relevant solvents. The initial part describes the retardation of fiber formation by CSF and they show that the effect levels off and is saturated at 50-60% CSF. The second part regards identification and quantification of the affected microscopic rate constants, underlying the observed retardation. Although I have some serious concerns on the kinetic analysis, the study breaks new ground and could be of general interest if some clarifications are made:

Response: We thank the reviewer for these positive words and we share his/her enthusiasm over the study.

Major concerns:

(i) My major concern regards the kinetic analysis, where the model used clearly do not apply for the aggregation kinetics in CSF. In buffer the fits are good, and the model seems to catch the overall macroscopic behaviour. However, for both CSF concentration (15% and 66%) the model seems to break down. The fitted functions do not at all fit the experimental data. Nevertheless the authors use the fitted microscopic rates to identify the mechanism for the aggregation retarding effect. The finding that the model is not applicable for A β 42 aggregation in CSF is indeed intriguing and important, but the authors oversees this, and simply uses the fitted values for their analysis. In my opinion, a detailed discussion, and preferably control experiments, on the reason for the model breakdown would strengthen the study significantly, and provide a line of further studies to understand the seemingly fundamental difference between A β 42 aggregation in buffer and CSF. The reproducibility within a CSF batch seem to be high, so the misfit should not be due to ambiguous data, but rather to that the aggregation mechanism alters and is dominated by steps not included in the model. The conclusions from comparing the fitted microscopic rate constants should be omitted, as they do not seem to report on the data.

Response: We agree with the reviewer that the fit is not 100% perfect, but the overall reason for the use of it was to see if primary nucleation and elongation only would fit the data, but it does not. We see a considerable improvement in the fits when adding a secondary process. To investigate whether this is true for other CSF concentrations, we have also added another CSF-series. We do think the fits catches the macroscopic behavior of the CSF series as well, especially considering the millions of components that constitutes CSF. Therefore, the slight deviations may be due to several different contributions from a number of these components; however, it will be more meaningful to tease out these contributions in the future with purified, single components.

We do thank the reviewer for his remarks, and have largely omitted the discussions regarding the microscopic rate constants, and focused mainly of the overall trend.

(ii) Also the seeding experiments show significant systematic deviations from the model. Maybe it would be better to just withdraw model free observables from the data, such as $t_{1/2}$ and maximum growth rate, to quantify the effect. Using the fitted values, again, do not report on the actual data, but merely on the fits. The beauty of the established model is that it captures the macroscopic characteristics and provides the microscopic rates, but in CSF, the model does not capture the macroscopic behaviour, and thus the obtained microscopic rate constants and saturation concentrations are very hard to interpret.

Response: We do see some deviation, and have therefore replaced the models in the manuscript with $t_{1/2}$ -plots to more clearly show the shorter fibrillation time. However, we still believe the fits show some very interesting differences between the primary nucleation and elongation and the addition of a secondary process, and so have kept the fits in the SI.

(iii) Third, the differences in A β 42 aggregation in the different CSF batches are interesting and could provide some additional information on the mechanism of the observed retardation. It would be of importance if the authors could characterise and report on the different CSF batches in terms of general properties, such as pH, ionic strength, overall protein concentration and trace metals. This could give a hint on the mechanism(s) that could be further tested.

Response: We thank the reviewer for this excellent idea. We have done a comparison of the two CSF pools we had available – one from Gothenburg and one from Lund – in terms of ionic strength (through conductivity), pH, and total protein concentration. We have added this in a table in the Supplemental Instructions.

Minor issue:

Overall, the figures (in the manuscript and supporting information) need to be edited for readability. Both the data legends and axis labels on inserts are too small.

Response: Thank you for observing this. We have now increased the font size of both the axis labels and the legends. On some we have clarified in the figure text, as the legend could not be increased.

Reviewer #2 (Remarks to the Author):

Frankel et al describes an aggregation kinetics study of Abeta42 in the absence and presence of human CSF. The authors performed several series of ThT kinetics assays in buffer with different Abeta42 monomer concentrations, in solutions containing varying concentrations of human CSF at constant monomer concentration, and in solutions containing 15% or 66.6% CSF with different Abeta42 monomer concentrations. The authors also reported cryo-TEM images of fibrils formed in CSF and seeded experiments using the fibrils generated in corresponding CSF conditions.

I completely agree with the complementary importance of the "top down" approach the authors employ here and I think studying aggregation kinetics in CSF is timely for the advancement of the field, and will help to bridge the gap between in vitro and in vivo knowledge. In fact, the observation by ThT assays and cryo-TEM reported here that fibrils with similar ThT growth traces and morphology can form in the presence of CSF compared with buffer is an important and satisfying result that validates previous research direction taken by the community using pure buffer to resolve the

molecular principles involved in Abeta and amyloid aggregation. The data reported in this manuscript is also of extremely high quality with the same high standards as previously published by the authors. The experiments are also very well documented. Therefore, overall I am very happy to see this manuscript, after revision, published so that the results and the data are made available to the community as soon as possible. My major comments, listed below, are mainly on the variability of the data and the model dependent interpretation of the data, and I very much hope the authors will find these useful in revising the manuscript.

Response: We thank the reviewer for this very positive note and we share his/her enthusiasm over the study.

– The authors report the use of four different pools of CSF. The authors also carefully documented and reported several different sets of ThT data, presumably repeated using different CSF pools. It is clear for example in supplementary Fig S4 that biological replicates of CSF do result in quite large variation in concentration dependent behaviour. Therefore, I think it will be very useful if the authors can label their datasets with the CSF pool number (e.g. as shown in Fig S4 simply with a number without breaching any ethical issues) so that data from the same pool can be linked through the manuscript.

Response: We agree that this is an excellent idea, and would clarify things. We have made additions in the text, and added a small hexagonal sign on each figure containing CSF, marked as to track the CSF pool used.

– The authors presented data with a constant 3uM concentration of Abeta42 and increasing concentrations of CSF. These data indicated an increase in the length of the lag phase as the concentration of CSF is increased. However, in later datasets this trend is apparently not preserved (e.g. the rate of the reactions in 15% CSF is faster than in 66% CSF). Is this a case of biological CSF pool variability? Since one of the conclusions is that the CSF may be giving a saturable inhibitory effect, I think it is important that the authors comment/clarify this. I understand the use of CSF may be very technically challenging, but if possible, I suggest the authors at least one more concentration of CSF to indicate what the actual trend is for the apparent saturation effect observed.

Response: We completely agree with the reviewer, as we have been considering this as well. To investigate, we did four series – i.e. we added a series with 32% CSF – with two different CSF pools from two different cities. We do observe the saturation effect for both these CSF pools, as the $t_{1/2}$ -plots of both 32% and 66% overlap closely.

– I understand completely the complexities in model fitting involved in the analysis of ThT data and I agree that the best fit reported by the authors give significantly better results than primary nucleation alone. However, it is still not easy for the readers to decipher the relative qualities of the models as well as other models not visually represented that the authors may have tested. For example, by eye some of the models presented in the supplementary data appear to fit better to the suggested model in the main text. Therefore I think the authors should report a table with some kind of quality of fit measure, assessment criteria, as well as the number of fitted parameters in the text or in the SI. Such inclusion would be extremely useful for the readers.

Response: We thank the reviewer for this nice idea. We have included a bar graph with the mean residual error from the global fits, which we used to compare the different models.

– In the text page 9, when discussing the saturation of secondary nucleation in pure buffer, the authors refer to figure 4A, which is not pure buffer but 15 % CSF according to the figure legend.

Response: We are very grateful to the reviewer for this careful scrutiny of our manuscript. It has now been revised.

– The top labels for Fig S8 is not consistent with the descriptions in the figure legend.

Response: As previously, it has now been revised.

– It is very interesting that the authors observe small particles that seem to be associated with the fibrils at higher concentrations. The authors suggest that they may be HDL-like components. If these particles are aggregates formed by Abeta42 from a competing pathway then that could potentially also contribute to the saturation effect the authors observe. Therefore, I think the discussions of the mechanistic origin of the saturation effect could be expanded in the discussions section to take into such possibilities.

Response: While we certainly think this is an interesting and noteworthy thought, we do not have any data that support the existence of competing pathway aggregates, so we do not feel comfortable adding such discussion. See also response to reviewer 3 below.

Reviewer #3 (Remarks to the Author):

The paper entitled: “Autocatalytic amplification of Alzheimer-associated A β 42 peptide aggregation in human CSF” is focused in the effect of CSF in amyloid aggregation kinetics. It is vastly known the lag of information about the amyloid aggregation of amyloid proteins in vivo. The main problems are the lag of reproducibility inter-assays and the low protein concentration present in transgenic animals. Thus, currently practically the unique possibility to track the amyloid aggregation in-vivo is in bacteria (in-cellulo assays) tracking the formation of inclusion bodies. In my opinion the paper is nice, interesting and well worked but I have two main constrains.

On one hand, I observe a discrepancy between figure 2 and the consecutive ones. In the figure 2 the authors show an increment of $t_{1/2}$ with CSF concentration (in brief the increment of CSF% increase the $t_{1/2}$). However in the other figures (see e.g. Fig6E), the authors show that 15% CSF inhibit amyloid aggregation more powerfully that 66% CSF using the same Ab42 range. I don't understand this. In addition the kinetics shown in figure 2 and the others display different kinetics (lag time, $t_{1/2}$) at similar Ab42 concentration and CSF%.

Response: We assume this might be due to CSF pool differences. We have now done four more series, with two different pools, as described previously and both pools give rise to the same trend.

On the other hand, although the authors only suggest (propose posterior studies) that effect of CSF is consequence of potential interactions with phospholipids and other compounds presents in CSF, the

unique evidence of this in the paper is the presence of several structures shown by TEM. I propose to soften these affirmations in this initial paper.

Response: We say in the manuscript: "Curiously, in several fields of the images, small particles seem to be associated with the surface of the fibrils and these features are more prevalent in images taken at the higher concentration of CSF. This is interesting in light of the identification of HDL-like components in the retarding fraction of size-separated CSF. However, the cryo-EM images alone do not allow for identification of the bound species and further investigation would be required for full disclosure." Therefore, we think that these affirmations are soft enough.

REVIEWERS' COMMENTS:

Reviewer #1 (Remarks to the Author):

In the revised version of the manuscript "Autocatalytic amplification of Alzheimer-associated A β 42 peptide aggregation in human CSF" the authors have answered and addressed most of my concerns, and the manuscript is, in my opinion, strengthened by the more conservative interpretation of the kinetic analysis, and I understand the arguing of the authors to include this analysis; to show that further processes than primary nucleation and elongation are necessary to comply with the observed kinetics.

Nonetheless, I still find it interesting that the model (that actually only reasonably good capture the fibrillation kinetics in buffer) further breaks down in CSF, both quantitatively and in overall features (the 'shape' of the graphs do not capture the shape of the kinetic traces). The paper would be much stronger if this very interesting result is discussed. However, this is an important and valuable discussion that could be raised, when the present findings are published and available for the scientific community.

I find the paper very interesting and important, on several layers, for the field and suggest publication in its present form.

Reviewer #2 (Remarks to the Author):

The authors have adressed all of my comments.

Reviewer #3 (Remarks to the Author):

The authors fully answer my suggestions. I propose this paper for be published in Comm Biol.

Reviewer #1 comment

In the revised version of the manuscript “Autocatalytic amplification of Alzheimer-associated A β 42 peptide aggregation in human CSF” the authors have answered and addressed most of my concerns, and the manuscript is, in my opinion, strengthened by the more conservative interpretation of the kinetic analysis, and I understand the arguing of the authors to include this analysis; to show that further processes than primary nucleation and elongation are necessary to comply with the observed kinetics. Nonetheless, I still find it interesting that the model (that actually only reasonably good capture the fibrillation kinetics in buffer) further breaks down in CSF, both quantitatively and in overall features (the ‘shape’ of the graphs do not capture the shape of the kinetic traces). The paper would be much stronger if this very interesting result is discussed. However, this is an important and valuable discussion that could be raised, when the present findings are published and available for the scientific community. I find the paper very interesting and important, on several layers, for the field and suggest publication in its present form.

We have added a new sentence in the discussion:
“The deviation in curve shape could possibly be caused by interactions in CSF between A β 42 and some species, with the dissociation of any formed complexes being rate limiting and affecting the reaction profile; however, further studies would be necessary for full disclosure of the possibilities behind the deviations.